# Risk Predictors of Advanced Fibrosis in Non-Alcoholic Fatty Liver Disease

**DOI:** 10.3390/diagnostics12092136

**Published:** 2022-09-02

**Authors:** Vasile-Andrei Olteanu, Gheorghe G. Balan, Oana Timofte, Cristina Gena Dascalu, Elena Gologan, Georgiana-Emanuela Gilca-Blanariu, Madalina-Maria Diac, Ion Sandu, Gabriela Stefanescu

**Affiliations:** 1Gastroenterology Department, Gr. T. Popa University of Medicine and Pharmacy, 16 Universitatii Street, 700115 Iasi, Romania; 2Institute of Gastroenterology and Hepatology, St. 1 Spiridon Emergency County Hospital, 1, Independentei Boulevard, 700111 Iasi, Romania; 3Department of Medical Informatics, Gr. T. Popa University of Medicine and Pharmacy, 16 Universitatii Street, 700115 Iasi, Romania; 4Forensic Sciences Department, Gr. T. Popa University of Medicine and Pharmacy, 16 Universitatii Street, 700115 Iasi, Romania; 5Institute of Legal Medicine Iasi, 700455 Iasi, Romania; 6Science Department, Interdisciplinary Research Institute, Alexandru Ioan Cuza University of Iasi, 11 Carol I Boulevard, 700506 Iasi, Romania; 7Academy of Romanian Scientists (AORS), 54 Splaiul Independenței St., Sector 5, 050094 Bucharest, Romania

**Keywords:** non-alcoholic fatty liver disease, advanced liver fibrosis, 2D shear wave elastography

## Abstract

The assessment of fibrosis in chronic liver diseases using non-invasive methods is an important topic in hepatology. The aim of this study is to identify patients with non-alcoholic fatty liver disease (NAFLD) and advanced liver fibrosis by establishing correlations between biological/ultrasound markers and non-invasively measured liver stiffness. This study enrolled 116 patients with non-alcoholic fatty liver disease, which were evaluated clinically, biologically, and by ultrasound. Liver fibrosis was quantified by measuring liver stiffness by shear wave elastography (SWE). Multiple correlation analysis of predictors of liver fibrosis identified a number of clinical, biological, and ultrasound parameters (BMI, blood glucose, albumin, platelet count, portal vein diameter, bipolar spleen diameter) that are associated with advanced liver fibrosis in patients with non-alcoholic fatty liver disease. The correlations between the degree of liver fibrosis and the risk values of some serological and ultrasound markers obtained in our study could be useful in clinical practice for the identification of advanced fibrosis in patients with NAFLD.

## 1. Introduction

Approximately 25% of the population is affected globally by various forms of non-alcoholic fatty liver disease (NAFLD), with a prevalence of up to 6.5% of the active form of disease represented by non-alcoholic steatohepatitis (NASH) [1]. The international scientific and professional community makes sustained efforts to stratify the risk of disease progression and establish a follow-up program for patients with non-alcoholic fatty liver disease [2]. The variability, invasiveness, and cost of the methods used to assess the grade of fibrosis in patients with NAFLD, are the elements that have prompted numerous research activities with the aim of identifying and implementing realistic, feasible, and reproducible fibrosis risk prediction strategies for this category of patients [2,3,4,5].

Research groups recommend that, in selected cases, non-invasive diagnostic methods, including imaging techniques and laboratory test markers, should be favored in determining the risk of progression of liver fibrosis [6,7,8,9,10].

We conducted a study to identify patients with NAFLD and advanced liver fibrosis by establishing correlations between laboratory tests/ultrasound markers and liver stiffness measured non-invasively by 2D-SWE.GE elastography.

## 2. Materials and Methods

We conducted a prospective study with diagnostic strategy, which took place in a tertiary gastroenterology and hepatology center, from January 2020 to June 2021. The research was conducted in a group of 116 consecutive patients aged > 18 years, diagnosed with NAFLD (history of at least 6 months), who agreed to participate in the study and undergo the proposed investigations.

Previously known patients with chronic liver disease of other etiologies, with previous splenectomy, pregnant women, and patients with hepato-portal encephalopathy were excluded.

The study was conducted after obtaining the approval of the Ethics Committee of ‘Grigore T. Popa’ University of Medicine and Pharmacy Iasi. Each patient included in the study signed an informed consent.

### Study Protocol

For each subject included in the study, personal pathological history, chronic medication, details of alcohol consumption, smoking, and dietary habits were noted. Clinical examination included determination of anthropometric indices as well as clinical examination. The paraclinical evaluation consisted of laboratory tests, ultrasound, and 2D-SWE.GE elastography.

Laboratory tests parameters included complete blood count, liver function tests, markers to exclude other etiologies of liver disease, cardiovascular risk parameters, lipid, carbohydrate, and protein metabolism balance.

All patients were examined fasting by abdominal ultrasound followed by 2D-SWE.GE elastography. Imaging explorations were performed by a single operator with a General Electric Logic 9, as shown in Figure 1 and Figure 2 (Figure 1 and Figure 2).

The ultrasound parameters assessed were: craniocaudal diameter of both, right lobe and left lobe of the liver (RLD and LLD), portal vein caliber (PV), and bipolar splenic diameter (BSD). The grade of steatosis was determined using a 4-grade classification system (grade 0—normal, grade 1—mild steatosis, grade 2—moderate steatosis, grade 3—severe steatosis) [11].

Elastometry was performed with the C1-6-D convex probe in Elasto 2D-SWE.GE mode. For liver stiffness (LS) expressed using Young’s modulus (kPa), the median value and IQR (interquartile range) were automatically calculated. The results were validated if the IQR/median ratio was less than 30%. The following cut-off values were used to qualify the grade of fibrosis: <6.6 KPa (F0–F1), 6.6–7.9 KPa (F2), 8–10 kPa (F3), >10.1 kPa (F4), values proposed by Kim et al. [12].

Subsequently, based on the threshold values of some serological markers, a logistic regression equation was determined to identify advanced liver fibrosis.

The statistical analysis of the data was performed in SPSS 27.0, using descriptive and inferential studies. Quantitative variables were characterized by descriptive statistics parameters and categorical variables by calculating the frequency distributions. The Shapiro–Wilk normality test was applied to determine which of the quantitative variables analyzed followed the law of normal distribution. In the case of variables that followed the normal distribution law, t-Student and ANOVA tests were used for comparisons, and the Mann–Whitney and Kruskal–Wallis tests were used for variables that did not follow the normal distribution law. In comparisons between more than two groups, we applied post hoc tests to locate statistically significant differences identified (LSD and Tamhane tests). The association between pairs of quantitative variables was assessed by calculating Pearson correlation coefficients and their associated level of significance. For the comparative study of the categorical variables, we used the chi-square test and estimated the risk factors by calculating the OR coefficients and associated confidence intervals, 95% CI. Risk factors were entered into a binary logistic regression model (forward LR method) to identify their relationship with the presence of advanced fibrosis diagnosis. The results obtained were considered statistically significant at *p* < 0.05.

## 3. Results

A total of 116 patients with non-alcoholic fatty liver disease were included in the study. Of these, 5 patients (4.31%) were excluded as no diagnostic values were obtained at the 2D-SWE.GE elastography assessment (IQR > 30%). One hundred and eleven patients diagnosed with non-alcoholic fatty liver disease, 62 males and 49 females aged between 26 and 76 years (mean age 53.56 years; *p* = 0.104) were fully assessed.

Mean values of the parameters investigated in the group of patients diagnosed with NAFLD are presented in Table 1.

### 3.1. Analysis of the Grade of Steatosis and Fibrosis in the Study Group

The distribution of steatosis grades in the NAFLD population was as follows: mild steatosis (S1) 5.4%, moderate steatosis (S2) 36.9%, severe steatosis (S3) 57.7%.

In relation to the distribution of liver stiffness values, the patients included in the study were divided into 3 subgroups: 50 patients with incipient fibrosis F0–F1 <6.6 kPa (45%), 30 patients with moderate fibrosis F2 = 6.6–7.9 kPa (27%), 31 patients with advanced fibrosis F3 + F4, >8 kPa (27.9%).

The distribution of fibrosis grades in relation to patient’s gender was as follows: male: F0–F1 43.9%, F2 25.6%, F3–F4 30.9%; female: F0–F1 48.3%, F2 31%, F3–F4 20.7%.

The distribution of fibrosis grades in relation to patients’ age was as follows: <40 years: F0–F1 52.5%, F2 32.5%, F3–F4 15%; 40–60 years: F0–F1 30.8%, F2 38.5%, F3–F4 30.8%; >60 years: F0–F1 46.7%, F2 15.6%, F3–F4 37.8%.

Regarding the relationship between the grade of fibrosis and the values of clinical and laboratory tests parameters assessed, statistically significant differences between the three categories of patients (F0–F1 vs. F2 vs. F3–F4) for the following parameters (Table 2): BMI (*p* = 0.000 *), glycaemia (9.015 *), albumin (*p* = 0.000 *), platelets (*p* = 0.000 *), RLD (*p* = 0.048 *), PV (*p* = 0.000 *), BSD (*p* = 0.000 *).

### 3.2. Analysis of Correlations between the Risk Values of Some Measured Parameters and Liver Stiffness

Table 3 shows Pearson correlation coefficients between parameters analyzed and liver stiffness (kPa).

For advanced F3−F4 fibrosis, inversely proportional correlations between advanced F3−F4 fibrosis and platelet count (r = −0.3450; *p* = 0.005 *) and albumin (r = −0.4161; *p* = 0.001 *) were found, respectively, directly proportional correlations between advanced F3−F4 fibrosis and glycaemia (r = 0.2984; *p* = 0.017 *), PV (r = 0.3612; *p* = 0.003 *), respectively, BSD (r = 0.3862; *p* = 0.002 *).

Figure 3, Figure 4, Figure 5, Figure 6 and Figure 7 suggest the correlations are statistically significant, related to correlation intensity and regression line between kPa (elastometry) and investigated variables.

### 3.3. Distribution of the Risk Values of the Parameters Analyzed

In the whole group of patients diagnosed with NAFLD, the highest frequencies of risk values were found for parameters: BMI (62.2%), cholesterol (59.5%), RLD (53.2%), GGT (49.5%), triglycerides (47.7%), BSD (40.5%), glycaemia (37.8%), and PV (38.7%), respectively.

The following are risk values for patients with NAFLD in relation to fibrosis grade (F0–F1 vs. F2 vs. F3–F4):for grade F0–F1, the most common risk values were for parameters: cholesterol (56%), GGT (44.0%), BMI (40.0%);for grade F2, the most common risk values were for parameters: BMI (76.7%), cholesterol (70%), triglycerides (60%), RLD (60%), GGT (46.7%);for grade F3–F4, the most common risk values were for parameters: BMI (83.9%), platelets (74.0%), RLD (71.0%), PV (71.0%), BSD (67.7%), GGT (61.3%), cholesterol (54.8%), triglycerides (54.8%).

### 3.4. Determination of a Logistic Regression Equation

The final objective of the study was to determine a logistic regression equation to diagnose advanced liver fibrosis in relation to the threshold values of non-invasive markers. In a first step, the risk values associated with each of the laboratory tests markers investigated for advanced liver fibrosis (F3–F4) were determined (Table 4).

The following risk factors have been identified for advanced fibrosis (F3–F4):BMI: OR = 4.474–high risk;Platelets: OR = 9.230–very high risk;RLD: OR = 2.841-high risk;PV: OR = 6.868–very high risk;BSD: OR = 4.900–high risk.

Table 5 shows predictors identified for advanced fibrosis-binary logistic regression.

Predictors identified for advanced fibrosis by multivariate analysis are (Table 5):Platelet counts below the risk threshold with a risk of 10.874:Dilated PV above the risk threshold, with an associated risk of 8.234:

In the last step, we determined the logistic regression equation:


*Logit (p) = ln(p/(1 − p) = −1.362 + 2.386 × Platelet counts + 2.108 × Modified PV (above 13 mm)*



*p = probability of developing F3–F4 fibrosis*


## 4. Discussion

Analyzing the results obtained, it is found that for most patients with NAFLD in the study group, a grade of severe steatosis S3 (57.7%) and significant fibrosis (54.9% moderate and severe fibrosis F2–F4) was documented. This distribution, with the prevalence of cases of increased severity, can be explained by the fact that the study was conducted in a group of patients who addressed a tertiary center. Most commonly, patients with mild steatosis and low fibrosis are seen on an outpatient basis in primary care.

Taking into account the demographic parameters, Leonardo et al. [13] conducted a meta-analysis which concluded that age and gender are major physiological factors at risk of developing NAFLD, along with race and genetic factors. In our study, the prevalence of male patients (73.8%) was found among patients diagnosed with non-alcoholic fatty liver disease, a result similar to that reported by Camhi et al. [14] and most studies in the literature. While in our study, a higher number of male patients with NAFLD was recorded in the whole group (regardless of the degree of fibrosis), there were no statistically significant differences between male and female patients reported in the subset of patients with severe steatosis (S3) and significant fibrosis (F2, F3, and F4) compared to Ciecko-Michalska [15], who reported a significantly higher prevalence of advanced fibrosis in male patients compared to female patients.

Another risk factor involved in the development and progression of NAFLD is age, with advanced fibrosis being significantly more common in older patients [1]. A number of studies that have analyzed groups of patients with non-alcoholic fatty liver disease reported a mean age between 51 years [16] and 63 years [17]. The mean age of patients with non-alcoholic fatty liver disease in our study was 53.56 years, which is within the range of values presented by the literature data.

Regarding the distribution of fibrosis grades by age category, there were statistically significant differences: in patients under 40 years of age cases with normal liver stiffness values are more common, while in patients over 60 years of age the incidence of cases with advanced fibrosis increases. These results are consistent with data in the literature [1,16].

BMI and metabolic parameters, along with demographic data, are major factors incriminated in the development of hepatic steatosis and progression to fibrosis. In our group of patients with NAFLD, 60.7% of patients with obesity (BMI above 30 kg/m^2^) were identified. A percentage of 59.5% of patients with risk values for serum cholesterol, 47.7% with risk values for triglycerides, and 37.8% of patients with hyperglycemia were also enrolled. In this regard, insulin resistance is currently considered to be a major mechanism in the development and evolution of non-alcoholic fatty liver disease towards steatohepatitis and advanced fibrosis [18,19]. In support of this hypothesis, there are also a number of results reported in the literature that show that the prevalence of non-alcoholic fatty liver disease can reach up to 80% in diabetic patients and advanced fibrosis is identified in a higher percentage in diabetic patients than in patients without diabetes [16,17,20,21].

A large number of serological markers have been proposed for the non-invasive assessment of liver fibrosis. In the particular case of NAFLD, it is considered necessary to differentiate tests based on serological markers that can be used in the diagnosis of steatosis from those that can be used in the prediction of severe fibrosis [10].

The relationship between advanced fibrosis and the risk values of clinical, laboratory tests, and ultrasonographic parameters was analyzed. The following distribution of liver stiffness values was recorded in our study group: <6.6 kPa—45.0%; 6.6–7.9 kPa—27.0%; 8–10 kPa—20.7%; >10.1 kPa—7.2%.

Analyzing the parameters measured from the perspective of frequency of risk values, in the subgroup of patients with NAFLD and early or moderate fibrosis, the most frequently recorded risk values (in more than one-third of patients) were found for parameters: BMI, cholesterol, glycemia, GGT, triglycerides, and ultrasound changes, respectively, RLD and BSD. For patients with NAFLD and advanced fibrosis, the same parameters identified in patients with mild but significantly higher fibrosis (over 50%) are found with increased frequency. In addition, thrombocytopenia and PV diameter are added. These findings are consistent with data in the literature [6,7,8,22,23,24].

The correlations between the values of the parameters followed and the numerical values obtained by elastometry were analyzed. In this context, the statistical analysis performed revealed the existence of directly proportional, statistically significant correlations of elastometric values with: BMI (moderate correlation), PV caliber (moderate correlation), and RLD (weak correlation). There was also a statistically significant inversely proportional correlation between platelet count and elastometric values. These results are fully consistent with data in the literature [25].

In relation to the relationship of cytolysis enzymes to elastometric values, no statistically significant correlations were identified. Results are contradictory in the literature: some studies have shown that elevated liver cytolysis enzymes are considered predictive for advanced fibrosis in NAFLD [26,27,28], and other studies have also shown that liver cytolysis levels do not correlate with the grade of steatosis or fibrosis [29].

Other parameters analyzed to identify correlations with liver stiffness were GGT, serum albumin, blood glucose, cholesterol, and triglycerides. For GGT values, the statistical analysis revealed the existence of a directly proportional, statistically significant correlation with elastometric values. Literature data show that in patients with non-alcoholic fatty liver disease GGT may be increased, with values of this parameter correlating with both advanced fibrosis and increased mortality [28]. For serum albumin values statistical analysis revealed a statistically significant, moderate inversely proportional correlation. These results are supported by the literature [30].

Serum triglyceride values correlate directly proportionally, statistically significant, but with low significance, to elastometric values, while serum cholesterol values correlate inversely proportionally, but statistically insignificant, to elastometric values.

In the studied group, glycemia (G) values were directly proportionally correlated with elastometric values. In most studies in the literature, diabetes mellitus is considered an important risk factor for hepatic steatosis, advanced fibrosis, and cirrhosis [17,20,29].

The study also aimed to identify a logistic regression equation to take into account the predictors for advanced fibrosis and to be easily used in the detection of advanced fibrosis.

Two predictors were identified for advanced fibrosis: the first predictor was platelet counts below the risk threshold and the second predictor was the PV diameter. Taking into account these identified predictors, using statistical analysis methods we obtained a logistic regression equation for advanced fibrosis exposed above.

Considering the specificity of the progression of the liver disease, a prospective study could not be carried out, because long monitoring intervals were necessary to evaluate the progression of liver fibrosis and the occurrence of clinically significant portal hypertension.

The advantage of the non-invasive character of the investigative methods used is realized from the perspective of the objectivity and sensitivity of the data obtained, in a real limitation represented by the lack of liver histopathological exploration as a control operator, with diagnostic value of certainty, regarding the real levels of fatty liver load and of liver fibrosis.

The real advantage of daily clinical practice is the stratification of the risk of developing liver fibrosis and clinically significant portal hypertension.

## 5. Conclusions

Following the implementation of mathematical modeling methods, a number of parameters can be used as tools to estimate the severity of chronic liver disease. Obesity, hyperglycemia, hypoalbuminemia, thrombocytopenia, portal vein dilation, and splenomegaly are predictive factors for advanced fibrosis, with the highest predictive power being platelet count and portal vein diameter.

The results on the correlations between the degree of liver fibrosis and the risk values of some serological and ultrasound markers obtained in our study could be applied in clinical practice, which are supported by the data in the literature.

The strength of the study is the use of non-invasive tests to evaluate liver fibrosis, considering the fact that there are few data in the literature about 2D-SWE.GE elastography. The limitation of the study is the reduced number of patients included and the absence of correlation with histological findings.

In order for the logistic regression equation resulting from the statistical analysis to prove its practical usefulness for the non-invasive and easy identification of advanced fibrosis, further studies are needed on an increased number of patients.

## Figures and Tables

**Figure 1 diagnostics-12-02136-f001:**
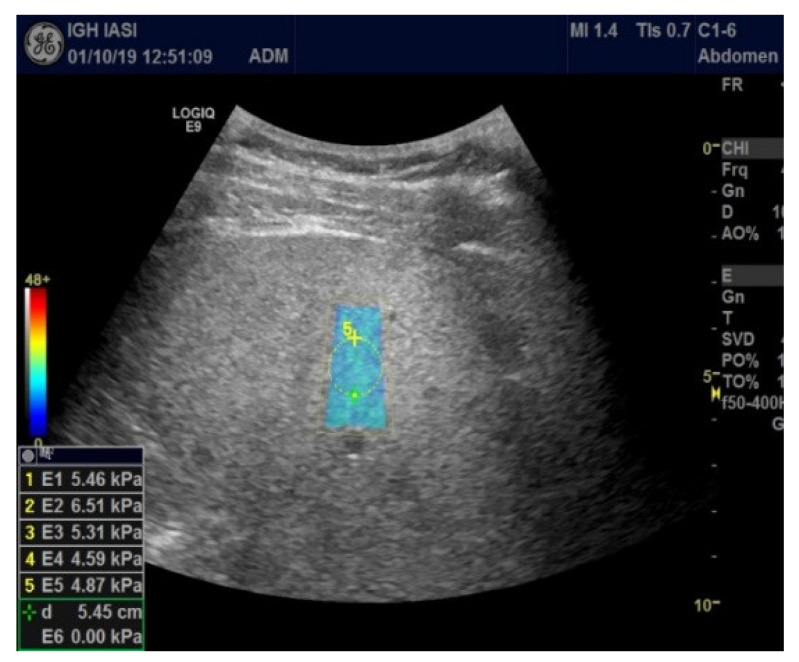
2D-SWE.GE elastography (measurement panel, color map).

**Figure 2 diagnostics-12-02136-f002:**
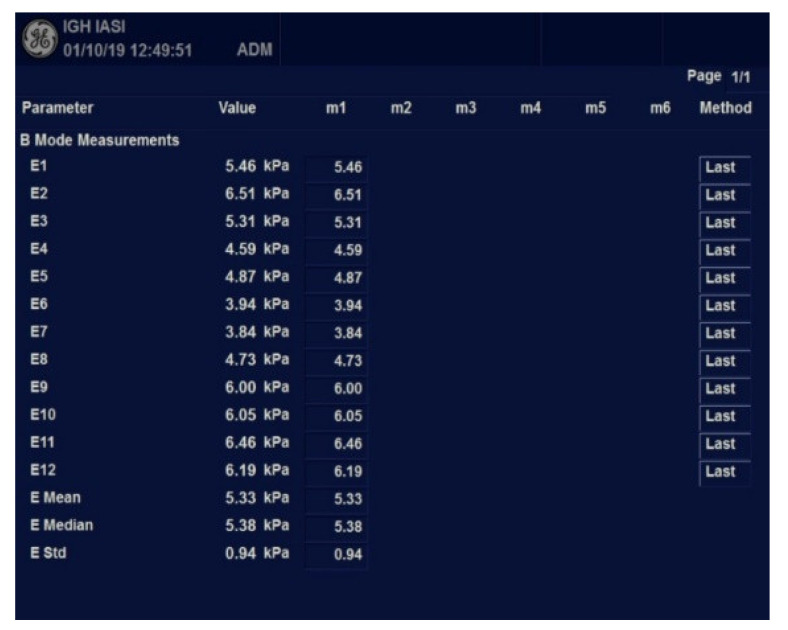
Worksheet with measurements results using 2D-SWE.GE.

**Figure 3 diagnostics-12-02136-f003:**
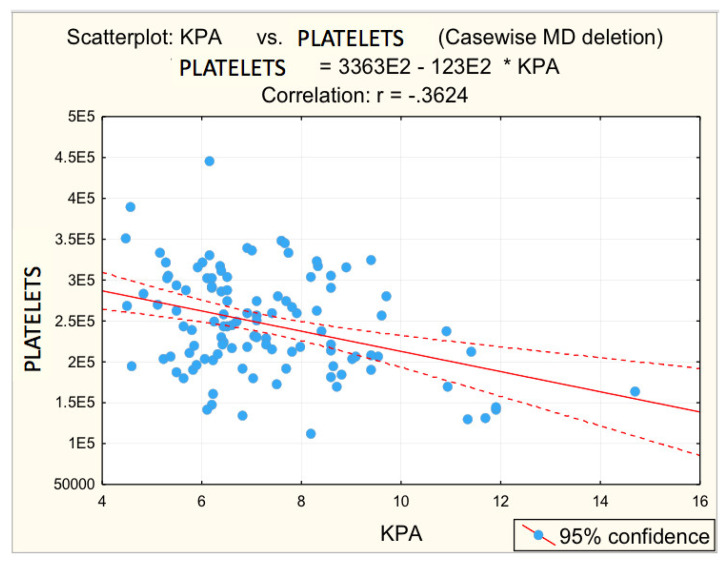
Correlation platelets—fibrosis. (E2 = 10^2^, * = multiplication, -.3624 = −0.3624).

**Figure 4 diagnostics-12-02136-f004:**
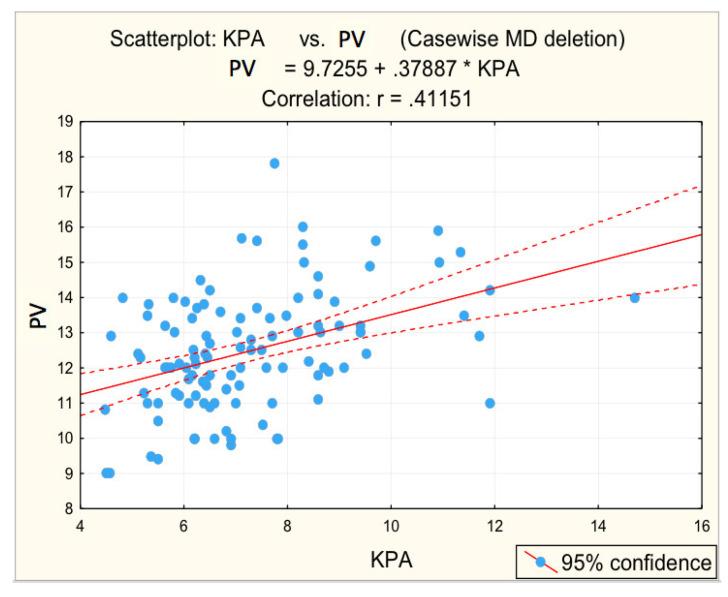
Correlation portal vein dimension—fibrosis.

**Figure 5 diagnostics-12-02136-f005:**
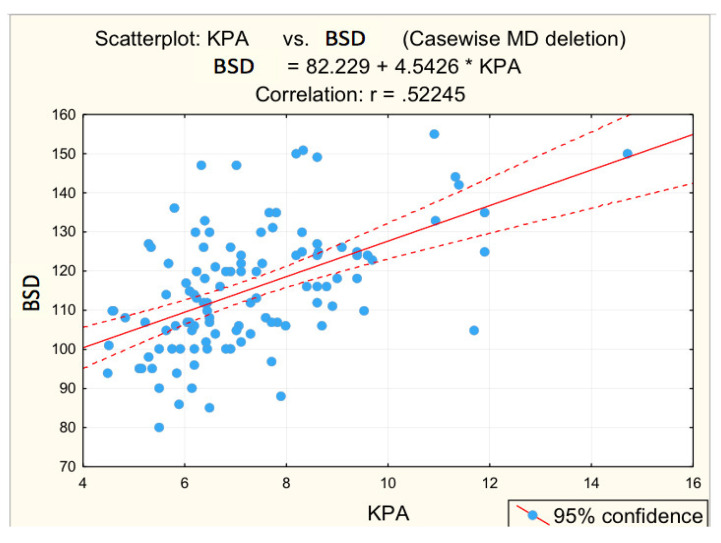
Correlation BSD—fibrosis.

**Figure 6 diagnostics-12-02136-f006:**
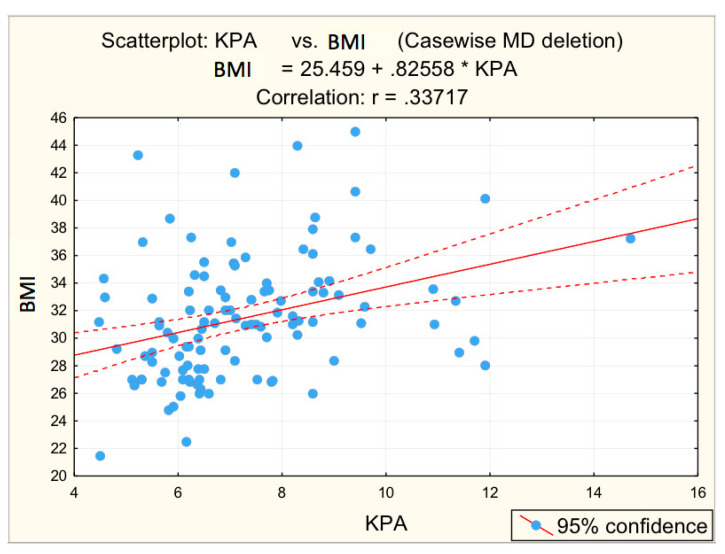
Correlation BMI—fibrosis.

**Figure 7 diagnostics-12-02136-f007:**
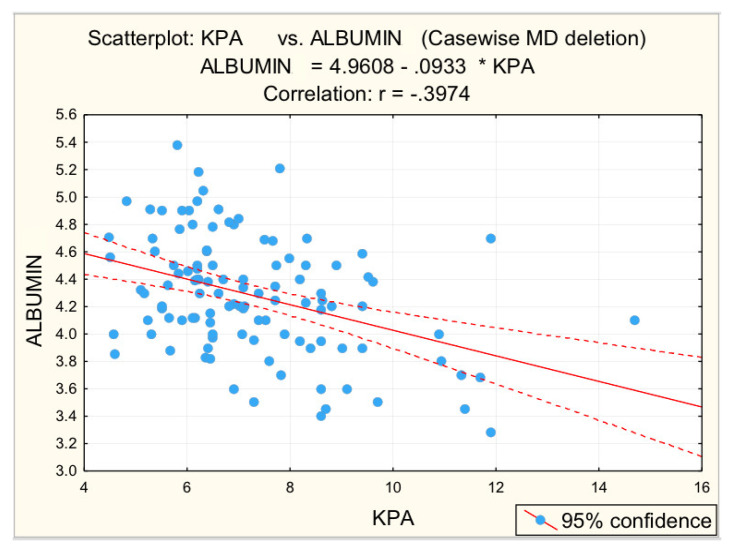
Correlation albumin—fibrosis.

**Table 1 diagnostics-12-02136-t001:** General characterization of the parameters analyzed.

Parameter	Mean Value	Standard Deviation	Standard Error	Min	Max	Shapiro–Wilk Test	
						W	*p*
BMI	31.466	4.4437	0.4218	21.500	45.000	0.96478	0.005 *
Platelets (/mm^3^)	246,450.450	61,862.860	5871.763	112,000	445,000	0.98723	0.377
ASAT (U/L)	26.850	14.799	1.405	10.000	128.000	0.682	0.000 *
ALAT (U/L)	37.54	23.694	2.249	6.000	112.000	0.847	0.000 *
ASAT/ALAT	0.8142	0.311	0.030	0.400	1.600	0.928	0.000 *
GGT (U/L)	57.14	47.179	4.478	13.000	334.000	0.747	0.000 *
Albumin (g/L)	4.2817	0.426	0.041	3.280	5.380	0.994	0.933
Blood glucose (mg/dL)	118.26	49.631	4.711	75.000	358.000	0.650	0.000 *
Serum cholesterol (mg/dL)	214.60	44.424	4.217	126.000	356.000	0.989	0.517
Serum triglycerides (mg/dL)	186.10	109.676	10.410	42.000	512.000	0.879	0.000 *
RLD (mm)	143.550	15.353	1.457	112.000	187.000	0.957	0.001 *
Portal vein caliber (mm)	12.482	1.671	0.159	9.000	17.800	0.988	0.397
BSD (mm)	115.280	15.780	1.498	80.000	155.000	0.983	0.135

* Statistically significant value (*p* < 0.05).

**Table 2 diagnostics-12-02136-t002:** Comparative study of parameters analyzed by grade of fibrosis.

Parameter	Mean ± SD				*p*
	Total	F0–F1	F2	F3–F4	
BMI	31.466 ± 4.444	29.670 ± 4.135	31.797 ± 3.444	34.042 ± 4.547	0.000 *
Platelets (/mm^3^)	246,450.45 ± 61,862.86	261,340.00 ± 61,821.537	248,200.00 ± 53,308.730	220,741.94 ± 63,145.318	0.000 *
ASAT (U/L)	26.850 ± 14.799	23.360 ± 6.375	28.100 ± 12.400	31.260 ± 23.368	0.312
ALAT (U/L)	37.540 ± 23.694	32.880 ± 20.239	40.230 ± 22.071	42.450 ± 29.158	0.122
ASAT/ALAT	0.814 ± 0.311	0.8526 ± 0.338	0.760 ± 0.262	0.8048 ± 0.312	0.551
GGT (U/L)	57.140 ± 47.179	48.240 ± 3.415	56.630 ± 41.536	71.970 ± 65.654	0.122
Albumin (g/L)	4.282 ± 0.4263	4.429 ± 0.385	4.3037 ± 0.398	4.0230 ± 0.406	0.000 *
Blood glucose (mg/dL)	118.260 ± 49.631	102.000 ± 17.131	100.100 ± 14.884	162.060 ± 74.735	0.015 *
Serum cholesterol (mg/dL)	214.600 ± 44.424	212.360 ± 38.017	224.330 ± 47.098	208.810 ± 50.927	0.354
Serum triglycerides (mg/dL)	186.100 ± 109.676	162.160 ± 90.569	198.770 ± 122.890	212.450 ± 119.482	0.128
RLD (mm)	143.550 ± 15.353	139.820 ± 14.245	144.970 ± 13.283	148.190 ± 17.742	0.048 *
Portal vein caliber (mm)	12.482 ± 1.6709	11.960 ± 1.374	12.237 ± 1.8922	13.561 ± 1.4049	0.000 *
BSD (mm)	115.280 ± 15.780	108.18 ± 14.026	114.930 ± 13.272	127.060 ± 13.921	0.000 *

* Statistically significant value (*p* < 0.05).

**Table 3 diagnostics-12-02136-t003:** Pearson correlation coefficients between parameters analyzed and liver stiffness.

Fibrosis (kPa) vs.	Total		F3–F4	
	r	*p*	r	*p*
BMI	0.337	0.000 *	0.1508	0.234
Platelets (/mm^3^)	−0.362	0.000 *	−0.3450	0.005 *
ASAT (U/L)	0.130	0.173	0.138	0.278
ALAT (U/L)	0.067	0.487	0.036	0.777
ASAT/ALAT	0.012	0.901	0.174	0.168
GGT (U/L)	0.203	0.033 *	0.234	0.063
Albumin (g/L)	−0.397	0.000 *	−0.416	0.001 *
Blood glucose (mg/dL)	0.442	0.000 *	0.2984	0.017 *
Serum cholesterol (mg/dL)	−0.103	0.281	−0.221	0.079
Serum triglycerides (mg/dL)	0.187	0.050 *	0.142	0.263
RLD (mm)	0.211	0.026 *	0.125	0.325
Portal vein caliber (mm)	0.412	0.000 *	0.3612	0.003 *
BSD (mm)	0.522	0.000 *	0.386	0.002 *

* Statistically significant value (*p* < 0.05).

**Table 4 diagnostics-12-02136-t004:** Risk values for advanced fibrosis (F3–F4).

Risk Values	Advanced Fibrosis				Pearson CHISQ	*p*	Odds Ratio	OR: 95% CI	
	YES		NO					Min	Max
	n	%	n	%					
BMI	26	83.9%	43	53.8%	8.619	0.003 *	4.474	1.561	12.827
Platelets (/mm^3^)	23	74.2%	19	23.8%	24.171	0.000 *	9.230	3.551	23.991
ASAT (U/L)	8	25.8%	10	12.5%	2.912	0.088	2.435	0.859	6.904
ALAT (U/L)	12	38.7%	24	30.0%	0.773	0.379	1.474	0.620	3.506
ASAT/ALAT	7	22.6%	19	23.8%	0.017	0.896	0.936	0.349	2.512
GGT (U/L)	19	61.3%	36	45.0%	2.372	0.124	1.935	0.830	4.511
Albumin (g/L)	4	12.9%	0	0.0%	0.005	0.005 *	-	-	-
Blood glucose (mg/dL)	5	16.1%	4	5.0%	3.714	0.054	3.654	0.912	14.642
Serum cholesterol (mg/dL)	17	54.8%	49	61.3%	0.381	0.537	0.768	0.332	1.776
Serum triglycerides (mg/dL)	17	54.8%	36	45.0%	0.867	0.352	1.484	0.645	3.415
RLD (mm)	22	71.0%	37	46.3%	5.482	0.019 *	2.841	1.165	6.928
PV (mm)	22	71.0%	21	26.3%	18.826	0.000 *	6.868	2.732	17.262
BSD (mm	21	67.7%	24	30.0%	13.203	0.000 *	4.900	2.008	11.956

* Statistically significant value (*p* < 0.05).

**Table 5 diagnostics-12-02136-t005:** Predictors identified for advanced fibrosis*-*binary logistic regression.

	B	S.E.	Wald	df	*p*	Exp (B)	95% C.I. for EXP(B)	
							Lower	Upper
Platelets	2.386	0.558	18.292	1	0.000 *	10.874	3.643	32.457
PV	2.108	0.555	14.406	1	0.000 *	8.234	2.772	24.456

* Statistically significant value (*p* < 0.05).

## Data Availability

All required data provided in the manuscript.

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
