# Peer review of "Risk Predictors of Advanced Fibrosis in Non-Alcoholic Fatty Liver Disease"

_diagnostics, 2022, doi:10.3390/diagnostics12092136_

Round 1
Reviewer 1 Report
Good job, nice presentation, important area of research.
Comments:
You should include section of abbreviations.
Better to include sample size in your statistical analysis section.
Line 203: what you mean by Modified PV in the equation ??? needs clarification.
No need to present discussion to sections but it needs to be in a single presention and you need to remove 4-1 and 4-2 in lines 248 and 260.
Line 268-270: In the literature, the results are contradictory: there are studies that have shown that elevated liver cytolysis enzymes are considered predictive for advanced fibrosis in NAFLD [26-28]. On the other hand, studies have also shown that liver cytolysis levels do not correlate with the grade of steatosis or fibrosis [29].
It better to be rephrased in a better way.
Line 279: These results are supported by the literature.. ..Where is the references???
You need to add a paragraph stress the limitation and the strength of the study.
Conclusion: needs rephrasing to better shape.
Author Response
Dear reviewer,
We appreciate for your precious time in reviewing our paper and providing valuable comments. It was your valuable and insightful comments that led to possible improvements in the current version.
- We included the section of abbreviations; we mentioned the sample size in the part of Materials and method, also in Results, so we did not included it in the statistic analysis for avoiding redundancy.
- Modified PV in the equation was referring to values of portal vein caliber superior to 13 mm, we added this information in the equation; thank you for pointing this out.
- We removed 4-1 and 4-2 in the part of discussion.
- We rephrased lines 268-270 and we added the reference to the sentence in line 279; thank you for pointing this out.
- We added a paragraph to stress the limitation and strength of the study.

Reviewer 2 Report
This is a nice study and like the authors suggested correlations between the degree of liver scarring and values of serological and ultrasound risk markers obtained could be useful in clinical practice.
The language is ok
The protocol and results are also ok
I have no further suggestions
Author Response
Dear reviewer,
Thank you for your comments.
We appreciate for your precious time in reviewing our paper and providing valuable comments.

Reviewer 3 Report
Thank you very much for this very interesting manuscript.
1. The main question addressed by the research is to identify patients with NAFLD and advanced liver fibrosis by establishing correlations between laboratory tests/ultrasound markers and liver stiffness measured non-invasively by 2D-SWE.GE elastography
2. I consider the topic quite original and relevant in the field as it address a specific gap in the field
3. This research would like to implement realistic, feasible and reproducible fibrosis risk prediction strategies for this category of patients
4. I have no specific improvements to suggest to the authors regarding the methodology or further controls to be considered
5. The conclusions are consistent with the evidence and arguments presented and they address the main question posed
6. I think that the table and figures are ok.
Author Response

(The authors gave the same response as above.)
